# Antioxidant and Antimicrobial Evaluations of *Moringa oleifera* Lam Leaves Extract and Isolated Compounds

**DOI:** 10.3390/molecules28020899

**Published:** 2023-01-16

**Authors:** Mmabatho Kgongoane Segwatibe, Sekelwa Cosa, Kokoette Bassey

**Affiliations:** 1Department of Pharmaceutical Sciences, School of Pharmacy, Sefako Makgatho Health Sciences University, Molotlegi Street, Ga-Rankuwa, Pretoria 0204, South Africa; 2Department of Biochemistry, Genetics and Microbiology, University of Pretoria, Private Bag X20, Hatfield, Pretoria 0028, South Africa

**Keywords:** antimicrobial, antioxidant, bioautography

## Abstract

*Moringa oleifera*, native to India, grows in tropical and subtropical regions around the world and has valuable pharmacological properties such as anti-asthmatic, anti-diabetic, anti-inflammatory, anti-infertility, anti-cancer, anti-microbial, antioxidant, and many more. The purpose of this study was to assess the free radical scavenging ability of two extracts and two pure compounds of *M. oleifera* Lam (hexane, ethanol, compound E3, and compound Ra) against reactive oxygen species, as well as their reducing power and antimicrobial activities. Bioautography antioxidant assay, 2,2-diphenyl-1-picrylhydrazyl (DPPH), hydrogen peroxide (H_2_O_2_) free radical scavenging, and iron (iii) (Fe^3+^ to Fe^2+^) chloride reducing power assays were used to assess the extracts’ qualitative and quantitative free radical scavenging activities. Furthermore, the extract and the compounds were tested against both Gram-positive and Gram-negative bacterial strains suspended in Mueller–Hinton Broth. The extracts and pure compounds showed noteworthy antioxidant potential, with positive compound bands in the Rf range of 0.05–0.89. DPPH), H_2_O_2_, and Fe^3+^ to Fe^2+^ reduction assays revealed that ethanol extract has a high antioxidant potential, followed by compound E3, compound Ra, and finally hexane extract. Using regression analysis, the half maximal inhibitory concentration (IC_50_) values for test and control samples were calculated. Compound Ra and ethanol exhibited high antioxidant activity at concentrations as low as ≈0.28 mg/mL in comparison with n-hexane extract, compound E3, ascorbic acid, and butylated hydroxytoluene standards. The radical scavenging activity of almost all *M. oleifera* plant extracts against DPPH was observed at 0.28 mg/mL; however, the highest activity was observed at the same concentration for ascorbic acid and butylated hydroxytoluene (BHT) with a low IC_50_ value of 0.08 mg/mL and compound Ra and ethanol with a low IC_50_ of 0.4 mg/mL, respectively. The extracts and pure compounds of *M. oleifera* have little to no antibacterial potential. *M. oleifera* extracts contain antioxidant agents efficient to alleviate degenerative conditions such as cancer and cardiovascular disease but have little activity against infectious diseases.

## 1. Introduction

Natural products derived from medicinal plants, whether as pure compounds or standardized extracts, offer countless prospects for new drug discovery. This is due to the unparalleled availability of chemical diversity in them. According to the World Health Organization (WHO), medicinal plants serve the primary healthcare needs of more than 80% of the world’s population [1,2,3,4]. In particular, countries such as China, India, Japan, Sri Lanka, Thailand, and Korea stand to gain the most from these plants [5,6,7]. Plant-based medicines are well-known for their reliability, accessibility, and affordability. These plants’ medical usefulness comes from their bioactive phytochemical components, which have defined physiological effects on the human body. Some of the most significant bioactive plant secondary metabolites are alkaloids, flavonoids, tannins, terpenoids, saponins, and phenolic compounds [8,9,10]. The biological function of antioxidants in medicinal plants is crucial. This is because oxidative stress, which is brought on by excessive free radicals, is linked to several degenerative conditions, such as cancer, ischemic heart disease, atherosclerosis, diabetes mellitus, and neurological illnesses. Since free radicals play a crucial physiological function in metabolic processes, it is necessary to maintain a homeostatic equilibrium between their presence and antioxidants [11,12]. One of the major public health concerns worldwide is infectious diseases caused by pathogens such as *Staphylococcus aureus*, *Escherichia coli*, *Pseudomonas aeruginosa*, *Neisseria gonorrhoeae*, and *Streptococcus pyogenes* [13,14]. The increased antibiotic resistance of pathogens has resulted in escalating mortality and morbidity rates [15]. The World Health Organization (WHO) estimates that each year infections caused by multidrug resistant (MDR) bacteria cause approximately 700,000 deaths globally, affecting people of all ages, including 200,000 newborns [16]. Thus, one possible treatment for these MDR bacteria is the use of natural products such as medicinal plants utilized by traditional healers in the management of diseases caused by microbial pathogens. Many foods, including fruits and vegetables, contain antioxidants. Plants and animals maintain sophisticated systems of many types of antioxidants, such as glutathione, vitamin C, vitamin A, and vitamin E, as well as enzymes such as catalase, superoxide dismutase, and different peroxides. Traditional herbal treatments and dietary foods were ancient peoples’ principal sources of antioxidants and antimicrobials, which protected them from free radical damage and antibiotic resistance [17]. Gram-positive bacteria (GPB) are non-spore producing, facultative anaerobic bacteria that can affect the respiratory tract, skin, and soft tissues [18]. *Streptococcus pyogenes* and *Staphylococcus aureus* cause prevalent infections such as furuncles, pneumonia, phlebitis, meningitis, urinary tract infections, pharyngitis, localized skin infections, rheumatic fever, rheumatic heart disease, and streptococcal toxic shock syndrome [19]. The Gram-negative bacteria of *Neisseria gonorrhoeae*, *Pseudomonas aeruginosa*, and *Escherichia coli* are non-spore forming bacteria that are known to cause erythematous exudate of pharynx, lung infections, and urinary tract infections, respectively [20,21,22]. This high burden of infectious diseases, along with the poor health systems found in most African nations, heightens the danger of antimicrobial resistance (AMR) spreading and its repercussions [23]. The insufficiency of effective treatments, pathogen resistance to antibiotics, and oxidative stress brought on by free radicals in the human body system have all led to the search for novel therapeutic alternatives from plants. A possible source of effective and safe antioxidant and antimicrobial natural products is the medicinal plants used by traditional healers and claimed to have potential for the management of these conditions. This is because the plants or other natural products’ derived antioxidants are rich in phenolics, flavonoids, phenolic acids, lignans, and stilbenes secondary metabolites. These compounds often serve as the plants’ defense mechanism against adverse effects including ultra violet radiation, temperature, and mechanical damage. In addition, by reacting directly with the oxidation products of fatty acids, phenolic compounds can prevent adverse changes from occurring in living organisms. Phenolic compounds also exhibit antimicrobial activity, causing the inhibition of microbial growth by interfering with the transport of nutrients that are important to their function [24]. One such medicinal plant used by traditional healers is *Moringa oleifera* Lam. It is a perennial softwood tree with low-quality timber and a straight, long trunk (10–12 m in height). *Moringa oleifera,* popularly called drumstick tree or horseradish tree, is widely researched for its plethora of pharmacological properties that are affiliated to the many bioactive compounds identified and in come cases isolated from the plant. Many studies have been carried out to investigate the medical significance of the bioactive compounds, possessing several biological activities such as antimicrobial, anti-inflammatory, and antioxidant. Even though nearly all the plant parts are useful to mankind and livestock, the leaves have found application as food and nutritional supplements, which are formulated and marketed in different dosage forms. However, the biological activities and the isolation of compounds from *Moringa oleifera* of the South African ecotype are poorly documented in the literature. As a result, the antioxidant and antimicrobial properties of hexane extract, ethanol extract, compound E3, and compound Ra are reported in this study.

## 2. Results

### 2.1. Dry Mass and Percentage Yield of the Extracts

The mass and the percentage yield of the dried hexane and ethanol extracts were recorded as 14.99 g (2.91%) and 8.59 g (1.72%), respectively, from 500 g plant powder.

### 2.2. Structural Characterization and Elucidation of Compound E1

#### 2.2.1. HPLC-PDA and UPLC-MS of Compound E1

The HPLC chromatogram of E1 revealed a single peak at Rt = 2.9 min for the compound, while high-resolution UPLC-MS analysis of E1 resulted in a single peak at Rt of 11.48 with *m*/*z* ratio of 383.2020 (M+1). In addition, the mass fragments of E1 obtained from the high-resultion UPLC-MS/MS at 383, 259, 183, and 129 conform with its structural major fragments at 259 (M-C_18_H_27_O), 183 (M-C_18_H_21_O_2_), and 129 (M-C_7_H_13_O_2_) (See Appendix A).

#### 2.2.2. One- and Two-Dimension NMR Analysis of E1

The carbon-13 NMR analysis of E1, conducted after filtering out the noise due to the experimental background, resulted in 25 clear carbon signals. These signals appeared at δ 173, 147, 139, 124, 123, 119, 114, 60, 50, 49, 39, 37, 34 33, 32, 31, 30, 29.6, 29.5, 28, 27, 24, 22, 19, and 14 ppm. As for the H-1 (proton) NMR of E1, the integration of the proton signals afforded a total of 35 hydrogens as potentially being an integral part of E1. The multiplicity and the coupling constant of the protons as well as the C-13 assignments are summarized in Table 1.

The distortionless enhancement by polarization transfer (DEPT) of E1 was useful in confirming the saturated carbons that were present in E2. It was deduced from the experiment that indeed E1 consisted of two methyl (-CH_3_) groups as positive signals, ten methylene (-CH_2_) groups as negative signals, and a further 5 methylene (-CH) groups as positive signals. The number of quaternary carbons, five in the case of E1, was not revealed in this experiment. The HSQC signals of E1 positively confirmed the number of protonated, and by extension the quaternary carbons, inherent in the structure of E1. The experiment produced the observation that E1 comprised 20 protonated and 5 quaternary carbons in agreement with the DEPT results. Following the NMR interpretation of the other experiments, especially the C-13, H-1, DEPTH, and HSQC NMR experiments of E1, it was evident that E1 was made up of different moieties. These different units include a saturated cyclic or aryl unit at between 14 and 50 ppm, an O-C unit at 60 ppm, an –OH-substituted phenyl unit, as well as a carbonyl carbon (C=O) at δ 147 and 173 ppm, respectively. To determine how the different units linked up to form the structure of E1, an HMBC experiment was performed. From the HMBC long-*J* couplings that translate to how the respective units in E1 are connected, the diagnostic HMBC correlations from the experiment were constructed and are depicted in Figure 1A. Based on the assigned HMBC correlation of E1, its skeletal structure was constructed and is displayed in Figure 1B. 

At this juncture, the proposed structure of E1 (Figure 2) and the name for E1 was elucidated as 10-hydroxy-1,3-dimethylchrysen-3-yl)-5-hydroxypentan-1-one with *m*/*z* ratio of 344.2664 calculated from C_25_H_36_O_3_. To verify the authenticity of the proposed structure, the calculated mass of E1 was compare to the high-resolution mass obtained from UPLC analysis of E1. As far as all the reported parameters are concerned, one can say that the proposed structure of E1 was true. This is because the calculated mass of E1 based on the proposed structure was in agreement with the UPLC-MS mass-to-charge ratio. In addition, the mass fragments of E1 obtained from the high-resultion UPLC-MS/MS at 383, 259, 183, and 129 conformed with its structural major fragments at 259 (M-C_18_H_27_O), 183 (M-C_18_H_21_O_2_), and 129 (M-C_7_H_13_O_2_).

### 2.3. Structural Characterization and Elucidation of Compound E3

#### 2.3.1. HPLC-PDA and UPLC-MS of Compound E3

The HPLC results of compound E3 afforded a single peak that resolved at a retention time (Rt) of 2.5 min in a total run time of 15 min. The purity of the peak was determined as 97% and is considered good for a compound that has been isolated from a natural product. Similar to the analysis of E1, a high-resolution UPLC-MS analysis of E3 afforded a single compound peak at Rt time of 13.93 min and a high-resolution *m*/*z* ratio of 347.2564. From the fragmentation pattern of E3, the major daughter ions were 67, 81, 95, 223, and 235. One could then infer that these daughter ions were typical for a loss of carboxylic acid (*m*/*z* = 45), a phenolic group (*m*/*z* = 95), or acetylic acid group (*m*/*z* = 58). These fragmentation groups also tended to highlight the fact that E3 contained such groups as part of its structure.

#### 2.3.2. One- and Two-Dimensional NMR Analysis of E3

From the C-13 experiment on E3, twenty (20) carbon signals were observed, thus implying that E3 consisted of 20 carbon atoms. Because C-20 compounds are usually commonly found in plants as fatty acids or their esters, E3 was thought to be one such compound. These carbon peaks were signaled at ^13^C NMR (101 MHz, CDCl_3_) δ 207.02, 173.98, 171, 147.09, 139.03, 124.47, 123.99, 119.09, 114.08, 60.16, 39.89, 34.42, 31.44, 30.94, 29.67, 28.96, 24.81, 22.70, 22.6, and 14.21 ppm. Based on the C-13 NMR interpretation, the carbon atoms between 14.0 and 30.0 were representative of saturated carbon groups or their derivatives. The carbon at 60 ppm was thought to be an *O*-substituted carbon atom, a methoxy group other electronegative or electron-rich atoms or group as present in E3. Whereas the carbons at 114.09, 119.09, 123.99, and 124.47 were typical of a phenyl ring moiety, those at 138 and 147.09 ppm suggested that E3 should have an *O*-substituted phenyl ring system. The carbons at 171.00, 173.98, and 207.02 were also indicative of three carbonyl carbon units and a possible presence of a carboxylic acid unit in agreement with the information from the HPLC-PDA and UPLC-MS. The number of hydrogens present in E3 was 24 in total. The carbon and proton signals of E3 as well as the multiplicity of these signals are summarized in Table 2. To determine the protonated carbons in E3, two-dimensional experiments were conducted. Among these experiments were the HSQC and the DEPT experiments.

From the DEPT experiments, one is able to tell the number of CH_3_, CH_2_, and CH atoms from the quaternary carbons. Whereas CH_3_ and the CH usually appear as positive signals and facing up, the CH_2_ carbons appeared as negative signals and facing down while the quaternary carbons were not featured in this experiment. From the DEPT experiment of E3, a saturated methyl group (CH_3_) was signaled at δ 14.21, 22.61, and 22.70 ppm. The methyl groups at 22.70 and 22.61 were thought to be further downfield because they were placed in a chemical environment that was electron-rich. As for the methylene (CH_2_) groups, four of them were present in E3 at δ 24.81 (CH_2_), 28.96 (CH_2_), 29.6 (CH_2_), and 30.94 (CH_2_) ppm. Whereas three saturated methylene groups appeared at δ 31.44 (CH), 34.42 (CH), and 39.89 (CH), there were two additional ones, δ 60.16 (O=CH) and 124.47 (Ph-CH) ppm, to amount to a total of five methylene (CH) groups in E3.

As mentioned earlier on, carbon atoms at 207.02, 173.98, 171.00, 147.09, 139.03, 123.99, 119.09, and 114.08 ppm did not appear on the DEPT experiment and are indicative of the quaternary carbons. Two-dimensional heteronuclear single quantum spectroscopy (HSQC) is a NMR experiment that is useful in determining all the protonated carbons present in an organic compound. In the compound E3, the experiment confirmed that it comprised a total of twelve protonated carbons at 124.47:7.38, 60.16:4.13, 39.89:2.20, 34.42:2.30, 31.44:2.19, 30.94:2.06, 29.68:2.02, 28.96:1.63, 24.81:1.30, 22.70:1.29, 22.63:1.27, and 14.21:0.87 and eight non-protonated carbons at 207.02, 173.98, 171.19, 147.68, 139.03, 123.99, 119.09, and 114.08 ppm, thus confirming the information obtained from the DEPT experiment. At this juncture, it must be re-emphasized that E3 consisted of an *O*-substituted phenyl, a carbonyl, a carboxylic, and a methyl as well as saturated hydrocarbon moieties. For the structure of E3 to be fully elucidated, it was imperative to establish how the different moieties of E3 were connected. The heteronuclear multiple quantum (HMBC) experiment is very useful in this regard because it reveals both long and short coupling connectivity of the different units that make a proposed structure. From the HMBC experiment on E3, there were key diagnostic signals that indicated the connectivity of the different moieties of E3. These signals occurred between methylic protons at the 0.80 ppm 1-*J* coupling to the phenanthrene C-3. In like manner, the phenanthrene H-2 protons at δ 0.88 ppm 3-*J* were coupled to the carbonyl carbon of the carboxylic acid moiety at 207.02 ppm. Another connection was that between the 6,7-diacetyl-5-hydroxyphenyl unit of E3 and the phenanathree-1carboxylic acid fragment through a *3J* coupling of H-4 proton at δ 1.29 to the 6,7-diacetone-5-hydroxyphenyl unit carbon at 147.09 ppm. The skeletal structure and the HMBC correlations of E3 are displayed in Figure 3A,B respectively. 

Collating all the chromatographic and spectrometric interpretations afforded the elucidation of E3 as 6,7-diacetyl-5-hydroxyphenyl-3-methylphenanthrene-1-carboxylic acid (Figure 4). In order to established if compounds of E3 type had been previously reported from plants or other natural products, a thorough literature search was carried out. The results of the search proved that analogues of E3 had indeed been identified in *Moringa oleifera* leaves in particular (Lin et al., 2019) [25]. The analogue of E3 identified in Indian *M. oleifera* leaves was identified as ajugaside A. Even though E3 and ajugaside A have a similar phenanthrene derivative, that is the 5-hydroxyl phenyl phenanthrene carboxylic acid (5-hydroxylphenyl-di-cyclohexane), there was, however, modification between both compounds. At C-1 of ajugaside, it appeared as though there would have been a bio-de-glycosylation of its 1-methylacetyl glucose to a carboxylic acid group detected in E3.

An alternative biomodification E3 is a bio-carboxylation using the acetyl glucose and rearrangement and migration of the methyl group from position C-1 in ajugaside to position C-3 in E3. Whereas there was another methyl group at C-4 of ajugaside that was absent in E3, we propose a second bio-de-glycosylation at C-6 of ajugaside and further rearrangement of 2-methylpropanol to a diacetyl moiety at positions 2′ and 4′ in E3. In terms of biological activities of both compounds, none is available in the literature for ajugaside. We herein report that E3 is characterized by antioxidant activity against the DPPH (1,1-diphenyl-2-picrylhydrazyl) radical with a DPPH free radical scavenging activity of IC_50_ of 0.67 mg/mL, better than ascorbic acid with IC_50_ of 0.88 mg/mL, but less than that of butylated hydroxyl toluene with IC_5O_ value 0.08 mg/mL in vitro. To the best of our knowledge, this is the first time an ajugaside-type compound has been reported from *Moringa oleifera* leaves of the South African ecotype.

### 2.4. Charaterization and Structural Elucidation of Isolated Compound Ra

#### 2.4.1. Structural Characterization and Elucidation of Compound Ra

The importance of absorbance maxima (ʎ_MAX_) values in structural elucidation of compounds has already been emphasized. As for the isolate Ra, it was 196 nm. Other possible absorbances of Ra appeared at 251 and 261 nm. The HPLC analysis of Ra resolved as the major peak at a retention time of 3 min. In order to further confirm the purity of the Ra, 1.0 mg/mL was analyzed using UPLC-MS in positive mode. The results obtained revealed a single peak at retention time of 14.05 min, whereas the molecular (*m*/*z* ratio) of Ra was 355.0740, implying that the actual molecular mass of Ra should be 356.0740 since the analysis was conducted in a positive mode. As for the fragmentation pattern of Ra, the major daughter ion was 266 and the other fragments were 281, 250, 207, 191, 147, 133, 89, and 73.

#### 2.4.2. One- and Two-Dimensional NMR Analysis of Ra

From the C-13 experiment on Ra, 24 carbon signals were observed, thus implying that Ra consist of 24 carbon atoms. These carbon peaks were signaled at ^13^C NMR (126 MHz, CDCl_3_) 124.48, 123.96, 119.10, 114.03, 68.32, 68.04, 34.86, 34.52, 33.79, 31.91, 31.42, 30.21, 30.02, 29.67, 29.63, 29.49, 29.33, 29.14, 28.94, 28.55, 27.07, 22.66, 18.81, and 14.06 ppm. Based on the C-13 NMR interpretation, the carbon atoms at δ 124.48, 123.96, 119.10, and 114.06 ppm suggested Ra consisted of two olefinic (double) bonds groups. In addition, carbons at δ 68.32 and 68.04 were signals typical of two *O*-substituted carbons as an integral part of Ra. Furthermore, the carbons at δ 34.86, 34.52, 33.79, and 31.91 indicated Ra was likely to have four methylene (-CH) units while those at δ 31.42, 30.21, 30.02, 29.67, 29.63, 29.49, 29.33, 29.14, 28.94, 28.55, 27.07, and 22.66 could either be protonated methylene (CH_2_) groups or quarternary carbons. Lastly, Ra was found to have two saturated methyl (-CH3) groups that resonated at δ 18.81 and 14.03 ppm. The proton NMR experiment on Ra revealed that the protons in Ra integrated to 36 protons in total. From the two-dimensional HSQC experiment it was deduced that Ra was characterized by 20 protonated and four quaternary carbons (Cq) at δ 119.90, 29.67, 30.21, and 123.96 ppm. Table 3 summarizes the carbon-13, proton, multiplicity, and coupling constants as well as the protonated and quarternary carbons characteristic of Ra. The diagnostic long *J*-coupling correlations from the HMBC experiment interpretation revealed the different moieties that made the structure of Ra and are displayed in Figure 5B, while the proposed carbon skeleton of the structure of Ra is represented by Figure 5A. From Figure 5B, the connection between the five-cyclic, otherwise called pentacyclic, ring system that made up Ra is displayed. The diagnostic 3*J* couplings include the H-12 proton at 7.52 and the C-11 and C-18 at 68.03 and 34.86 ppm. This indicates the linkage between lower (ring AB) and upper bicyclic (DE) systems by ring C.

Combining the one-dimensional with the two-dimensional experiments (COSY, DEPTH, HSQC, and HMBC) as well as the UPLC-MS data of Ra, its structure (Figure 6), was elucidated as hexademethylated 3β,11β-dihydroxyfriedelane. This compound is a pentacyclic triterpene of the type isolated and reported from *Maytenus robusta* by Sousa and co-workers in 2012 [26] and by Salimi et al. (2019) [27] from Indonesian *Moringa oleifera* Lam. leaves’ hexane extract.

Comparing the structure of Ra and its analogue 3β,11β-dihydroxyfriedelane revealed the major differences between the two structures. Whereas Ra consisted of two methyl groups, 3β,11β-dihydroxyfriedelane had six additional methyl groups. It is also logical to say that the C-5 and C-12 methyl groups in 3β,11β-dihydroxyfriedelane could have been biosynthetically converted to the two olefinic groups in Ra. With such similarities between these compounds, we propose that Ra may be called hexademethylated 3β,11β-dihydroxyfriedelane. With a proposed molecular formula of C_24_H_36_O_2_ and a calculated molecular mass of 356.27, the high-resolution mass of 356.0740 obtained from UPLC-MS analysis of Ra hereby confirms the proposed structure. Furthermore, the fragmentation patterns of 281, 266, 250, 207, 191, 147, 133, 89, and 73 corresponded to (M-C_20_H_31_O), (M-C_17_H_30_O_2_), (M-C_16_H_25_O_2_), (M-C_13_H_21_O_2_), (C_12_H_19_O_2_), (C_11_H_16_), (C_10_H^17^), C_5_H_13_O), and (C_4_H_9_O), respectively, as the daughter ions in the compound.

### 2.5. Qualitative Antimicrobial Assay

An in vitro qualitative antimicrobial assay was performed only for compounds Ra and E3 due to insufficiency of E1. The antimicrobial activity screening was determined for the four extracts and pure compounds. The screening revealed that *S. aureus*, *S. pyogenes*, *P. aeruginosa*, *E. coli*, and *N. gonorrhoeae* were all resistant to the extracts. Ojiako (2014) [28] and Okorondu et al. (2013) [29] previously found that *M. oleifera* leaf extracts were potent against *E. coli* and *S. aureus* when extracted with both the non-polar solvent hexane and the polar solvent ethanol. However, no action against the organisms was found in the current investigation. This could be due to a variety of reasons, including genetic background and the concentration of the extract utilized. In addition, Semenya et al. (2020) [30] conducted a similar investigation that corroborated the findings of this study. The presence of bioactive secondary metabolites on the developed chromatograms was detected by the thin-layer chromatography (TLC) agar overlay bioautography assay. Contamination is a disadvantage of antimicrobial secondary metabolite detection. The agar overlay TLC–bioautography assay is the most dependable, cost-effective, simple, and sensitive assay available, with the added benefit of detecting antimicrobial metabolites in microbial extracts that are viable against bacteria and fungi. The colored background of formazan is produced in the agar overlay TLC–bioautography due to the dehydrogenase activity of microorganisms that converts vital dyes into a chromogenic product by the reduction process. The extracts and isolated compounds had no activity and therefore the plates had a purple color zone which denoted that there was no inhibition of test pathogens [31].

### 2.6. Qualitative Antioxidant Evaluation

*M. oleifera* leaf extracts were tested against the stable DPPH (2,2-diphenyl-1-picryl-hydrazyl-hydrate) free-radical technique to assess antioxidant activity. In these investigations, the ability to scavenge DPPH radicals was assessed by the staining of the solution. DPPH is a free radical that generates a violet solution in methanol and is stable at room temperature. When a free radical interacts with an antioxidant, it loses its free radical property and turns light yellow because of the chain breaking. The extracts that exhibited yellow creamy bands against the purple background in the DPPH free radical scavenging capacity experiment by TLC were regarded as having antioxidant potential. Antioxidants showed as yellow creamy bands on a light purple background because of this procedure [32,33]. The extracts of *M. oleifera* leaves (n-hexane, dichloromethane (DCM), EtOAc, and EtOH) were spotted and developed using various solvent systems with different ratios, including n-hexane: DCM: EtOAc (6:2:0.5 *w*/*v*); n-hexane: DCM: EtOAc (6:2:1 *v/v/v*); n-hexane: DCM: EtOAc (6:2:2 *v/v/v*); n-hexane: DCM: EtOAc (6:2:3 *v/v/v*); acetic acid: n-hexane (1:9 *v*/*v*) and n-hexane: DCM: EtOAc (6:2:0.8 *v/v/v*). The purpose of experimenting with different solvent systems was to achieve better resolution of the various antioxidant compounds in the four extracts. The dried developed plates were sprayed with 0.2% solution of DPPH. Figure 7 shows the difference in the number of substances extracted from the leaves of *M. oleifera* and separated by TLC analysis to observe antioxidant compounds. The hexane extract had as many as six antioxidant bands at Rf of 0.01–0.75. In addition, the DCM extract displayed three antioxidant bands at Rf of 0.58–0.75, thus highlighting it contained similar antioxidant compounds to those present in the hexane extract. On the other hand, the polar EtOAc extract revealed one band with potential antioxidant activity at Rf of 0.78. However, based on the very slight intensity of the cream stain against the purple background, one can say that the compound in the EtOAC extract had minimal antioxidant activity compared to the non-polar DCM and hexane extract. As for the EtOH extract, a single band at Rf value of 0.05 indicated strong antioxidant potential. Whereas the spot on the EtOH appeared as a dot plot, due to the mobile phase Hex:DCM:EtOAc (6:2:0.8 *v/v/v*) used for the analysis, we propose that the bands could appear more if a more solvent was used as the mobile phase. Even though the dichloromethane extract indicated the highest quantity of antioxidant constituents from *M. oleifera* leaves, herbal decoction of the plant prepared with a consumable alcohol such as ethanol should furnish the body of the user with antioxidants.

### 2.7. Quantitative Antioxidant

#### 2.7.1. DPPH Free Radical Scavenging Assay

The reaction between the DPPH and the plant extracts or isolated compounds was tested in 96-well plates. Yellow coloration of the solutions in the plates (Figure 8) was an indication of positive good radical scavenging potential of the plant extracts and isolated compounds E3 and Ra. DPPH is a stable free radical that accepts an electron or a hydrogen radical to form a stable diamagnetic molecule that is widely used in research on radical scavenging activity. In the DPPH radical scavenging assay, antioxidants react with DPPH (deep violet color) to produce yellow colored α, α-diphenyl-β-picryl hydrazine. The degree of discoloration indicates strong ability of the extract or isolated compound to scavenge free radicals [34].

It is of paramount importance to know which extracts or compounds from extracts have the best free radical scavenging activity, both nutritionally and clinically. Furthermore, the concentration of a plant extract or phytochemical(s) required to inhibit or scavenge 50% of the free radical in vitro is a useful tool when developing herbal medicines. Figure 8 depicts *M. oleifera* Lam’s DPPH radical scavenging activity. The extracts and pure compounds were able to neutralize the DPPH free radicals by donating hydrogen to a specific extent. As can be seen in Figure 8, the ethanol extract revealed an optimal antioxidant potential at a concentration of approximately 0.30 mg/mL followed by that of compounds Ra and E3 and lastly, the dichloromethane extract. Because it contains more phenolic compounds, ethanol inhibits free radicals more effectively than n-hexane. The decreasing DPPH absorbance of the test samples and controls were translated to percentage DPPH free radical scavenging (% DPPH antioxidant), as shown in Figure 8 above. Using regression analysis, the IC_50_ values (Table 1) for test and control samples were calculated. Compound Ra and ethanol exhibited high antioxidant activity at concentrations as low as ≈0.28 mg/mL in comparison with n-hexane extract, compound E3, ascorbic acid, and butylated hydroxy toluene standards. The radical scavenging activity of almost all *M. oleifera* plant extracts against DPPH was observed at 0.28 mg/mL; however, the highest activity was observed at the same concentration for ascorbic acid and BHT with a low IC_50_ value of 0.08 mg/mL, and compound Ra and ethanol with a low IC_50_ of 0.4 mg/mL, respectively. As illustrated in Figure 8, DPPH scavenging increased in a concentration-dependent manner.

#### 2.7.2. Hydrogen Peroxide Free Radical Scavenging Assay

Hydrogen peroxide (H_2_O_2_) can penetrate cellular membranes; as a result, H_2_O_2_ is extremely important in cellular metabolism. H_2_O_2_ is not particularly reactive, but it can be hazardous to cells when it produces hydroxyl radicals (OH^−^) in the cell. In the presence of oxygen, the lipid radical will launch a chain reaction, resulting in lipid peroxide, which then breaks down to malondialdehyde aldehydes [31]. In this study, the maximum hydrogen peroxide scavenging activity was found at 0.28 mg/mL concentration with 82% scavenging activity for the ethanol extract and E3. However, Ra indicated a relatively better scavenging activity at a concentration of 0.42 mg/mL at 89%, while the hexane extract exhibited the least activity in agreement with the results obtained for the DPPH assay. The IC_50_ values for the H_2_O_2_ assay (Table 1) were also in agreement with the trend in the free radical scavenging activity observed in the two assays (Figure 9).

#### 2.7.3. Ferric Reducing Power Assay

The reducing power activity of the extracts was determined by their capacity to contribute electrons to enable the reduction of ferric ions (Fe^3+)^ to ferrous ions (Fe^2+^). The absorbance at 700 nm was determined after different quantities of sample extracts were charged with Fe^3+^ solutions. This absorbance reflects the quantity of Fe^2+^ in solution; therefore, the greater the absorbance, the higher the concentration of Fe^2+^ and the capacity of the analyte to donate electrons; i.e., the higher the extract’s reducing power. The stronger the antioxidant activity, the greater the reducing power. The reducing power of *M. oleifera* leaves extracts and pure components are displayed in Figure 10.

As evident in Figure 10, the antioxidant activity of all the extracted and isolated compounds peaked at the same concentration range of 0.28 mg/mL, like that in the DPPH and free H_2_O_2_ assays. However, a slight deviation was observed for the hexane extract that matched that of the other extracts and isolated compounds. In addition, all the extracts and isolated compounds performed better than the ascorbic acid and BHT, which were employed as reference medicines. Overall, our findings indicate that *M. oleifera* leaves extracts and isolated compounds have potential as a natural antioxidant, as previously described [35]. As a result, the plant extracts may be effective in the treatment of oxidative stress-related ailments such as atherosclerosis, chronic obstructive pulmonary disease, Alzheimer’s disease, and cancer [36].

### 2.8. Statistical Analysis

The IC_50_ results presented in Table 4 were statistically analyzed to compare the significance of the antioxidant assay methods. The results obtained indicated that IC_50_ values of the DPPH and FRAP assays were significant with a *p*-value of 0.03.

## 3. Discussion

The plant leaves’ ethanol extract showed strong scavenging of DPPH at 65%, hydrogen peroxide with a percentage inhibition of 89%, and reducing power with a percentage inhibition of more than 85%. In some cases, the extracts showed the lowest percentage inhibition of any experiment compared to the standards (ascorbic acid and BHT). This does not preclude the plant leaves from being used as an antioxidant substitute because activity exists, albeit at a lower level than expected. Factors such as plant location or storage, drying procedure, solvent polarity (methanol instead of ethanol), and the contribution of carbohydrates in the extracts could have influenced the results. Ascorbic acid showed more activity against DPPH compared to ethanol extract, which has high polarity. In general, the plant leaves exhibited concentration-dependent but significant scavenging activity at the least concentration of 0.28 mg/mL. The results for both DPPH and hydrogen peroxide are motivating because traditional healers use water or alcohol to prepare decoctions from leaves and other parts. The observation that the plant leaves’ ethanol extract exhibited the best free radical activity agrees with a previous report by Nobossé and co-workers (2018) [37]. This group also found that ethanolic extract had the highest DPPH scavenging activity for *Moringa* leaves from Cameroon. The agreement between our results and those of other researchers is beneficial to South Africans in the area of solvent of choice for the best extraction of antioxidants from *M. oleifera*. Furthermore, it demonstrates that traditional healers will extract more antioxidants using alcohol, which can scavenge both DPPH and hydrogen peroxide. This plant has little potential to be the source of antibacterial agents that could be used to manage bacterial infections. The isolated compounds E3 and Ra, now known as 6,7-diacetyl-5-hydroxyphenyl-3-methylphenanthrene-1-carboxylic acid and hexademethylated 3β,11β-dihydroxyfriedelane, respectively, equally indicated concentration-dependent antioxidant activity. This implies that these compounds could serve as lead compounds that can be synthesized or form part of a library of compounds that may be developed into antioxidant drugs that could replace those that have not been as effective hitherto. The main limitation to this study was the low concentration of 1 mg/mL of extraction solution that was used to investigate the antimicrobial potentials of the extracts. This we thought was the reason we could not confirm activity for the extracts consistent with some other reports.

## 4. Materials and Methods

### 4.1. Sample Collection and Preparation

*Moringa oleifera* Lam leaves were collected from Ga-Mphahlele, South Africa, Limpopo province using a convenience sampling method. The leaves were collected in bulk and air-dried at room temperature, then ground into powder using an electrical grinder and stored in the laboratory cupboard until used. All solvents used for the extractions were of analytical grade (AR) and were purchased from Rochelle Chemicals, South Africa. Five hundred grams (500 g) of the finely ground plant material was mixed with 2300 mL of dichloromethane in 5000 mL Erlenmeyer flasks. The flasks were placed on an orbital shaker and the flasks and contents were shaken for 18 h to allow for extraction. The supernatant post-extraction was filtered using Whatman no. 1 filter paper into a round bottom flask and the process was repeated two more times. The filtrates were combined and concentrated using a Stuart rotary evaporator (RE400, Cole-Parmer Ltd. Stones, St15 OSA, UK), to afford the dried dichloromethane extract. The same procedure was followed using the same plant residues with 2300 mL of ethanol to give dried ethanol extract. The mass and percentage yield of each extract was determined using standard protocols.

### 4.2. Isolation of Compounds from the Hexane Extract

The plant hexane extract was prepared for column chromatography purification by dissolving 15 g of hexane extract in 100 mL of DCM. The solution was adsorbed with 30 g of dry silica using pestle and mortar and the solvent was allowed to evaporate at room temperature under a stream of air for 20 min. While the mixture was drying, 80.619 g of silica was mixed with ethyl acetate to form a homogenous slurry and stirred to eliminate bubbles using a glass stirring rod. The slurry was poured into a sintered glass column (C.C. Imenmman PTY, Johannesburg, South Africa) whose outlet narrowly opened to allow for packing of the silica gel. The adsorbed hexane extract-dry silica mixture was carefully loaded on the column gel bed. Cotton wool was placed onto the extract sample to avoid splashing when adding the mobile phase. The column was eluted with n-hexane: dichloromethane (6:2 *v*/*v*; 6: 2: 0.5 *v/v/v* and 6:2:1 *v/v/v*) to obtain a total of 530 fractions. The different fractions were analyzed by TLC, and the compound spots were visualized under the UV light at 254 and 365 nm and then derivatized with a combination of methanol and sulfuric acid (9:1 *v*/*v*) to expose those compounds that could not be seen with the naked eye. The fractions that had same retardation factor (Rf) value were bulked together, leading to 26 (A–Z) fractions. These major fractions were dried in pre-weighed beakers and their masses determined after air-drying in a fume hood.

#### 4.2.1. Bulking of Fractions from Hexane Extract Column

Contents of test tubes (TT) 1–14 were combined to result in major fraction A, since the compound bands on the TLC plate displayed similar Rf values. In like manner, contents of test tubes 15–40 were combined to afford fraction B, 41–90 afforded fraction C. Other fractions were pooled as follows respectively D (TT91-116), E (TT117-160), F (TT161-172), G (TT173-286), H (TT287-340), I (TT341-263), J (TT364-367), K (TT368), L (TT369-379), M (TT380-387), N (TT388-405), O (TT406-415), P (TT416-423), Q (TT424-437), R (TT438-439), S (TT440-450), T (TT451-473), U (TT474-485), V (TT486-495), W (TT496-500), X (TT501-509), Y (TT510-519), Z (TT520-530).

#### 4.2.2. Re-Chromatography of Major Fraction A

Fraction A was re-chromatographed because its TLC profile was semi-pure with a single but not compact spot. Silica gel 60 (160.02 g) was mixed with ethyl acetate using a stirring rod, and loaded into a glass column. The solvent was allowed to flow out so that the silica gel settled and formed a column bed. Then, 3.75 g of A was dissolved with 40 mL of DCM and loaded onto the gel in the column. The loaded column was eluted using n-hexane: DCM (6: 1 *v*/*v*). Twenty fractions were collected and concentrated using a rotary evaporator. These fractions were again analyzed using TLC and those that had similar bands with identical Rf values were combined together. Detailed bulking involved the combining of contents of test tubes 1 and 2 to give sub-fraction A_1_; contents of test tubes 3–16 were pooled together to yield sub-fraction fraction A_2_. Lastly, contents of test tubes 17–20 were combined together to form sub-fraction fraction A_3_. Since sub-fractions A_1_ and A_3_ indicated good potential to contain a pure compound because they displayed a single compact spot from TLC analysis; they were stored in the fridge for characterization. On the other hand, sub-fraction A_2_ was stored safely for further purification because it lacked the characteristics of sub-fraction A_1_ and A_3_.

#### 4.2.3. Re-Chromatography of Major Fraction A_2_

A glass column of was used for re-purification of sub-fraction A_2_. The column was prepared as earlier described. The sample was prepared by adsorbing 3.01 g of sub-fraction A_2_ with 65.0 g of silica gel. The mixture was dried in room temperature for 15 min. The dried sample was loaded carefully onto the silica gel slurry bed and eluted with pure n-hexane. One hundred and six (106) fractions were collected, concentrated and analyzed using TLC. These sub-fractions were bulked as follows: A_2.1_ (TT1-4), A_2.2_ (TT5-20), A_2.3_ (TT21-36), A_2.4_ (TT37-45), A_2.5_ (TT46-60), A_2.6_ (TT61-70), and A_2.7_ (TT71-106). Based on TLC profile obtained, A2.2 was further re-purified to A2.2.1 to A2.2.7 and upon re-crystallization of A2.2.1, Ra and Rb were realized with Ra indicating the best degree of purity.

### 4.3. Isolation of Compounds from the Ethanol Extract

The plant ethanol extract was prepared for column chromatography purification by dissolving 7.01 g of ethanol extract in 100 mL of ethanol. The solution was adsorbed with 80.03 g of silica gel using a pestle and mortar and the solvent was allowed to evaporate at room temperature under a stream of air for 25 min. While the mixture was drying, silica gel (160.0 g) was mixed with n-hexane to form a homogenous slurry and stirred to eliminate air bubbles using a glass stirring rod. The slurry was poured into a sintered glass column and the solvent was allowed to flow out of the column opening so that the gel could settle. The dried silica gel extract mixture was carefully treated and applied on the column gel bed. The mobile phase used for eluting the column consisted of hexane:DCM:EtOAc (6:2:0.5; 6:2:0.8; 6:2:1.5 *v/v/v*), DCM:EtOAC (8:2; 7:3; 6:4; 1:1; 4:6; 3:7; 2:8 *v*/*v*), EtOAc, and EtOAc:EtOH (8:2 *v*/*v*). A total of 880 fractions were collected and concentrated using a rotary evaporator. These fractions were analyzed by TLC, and the compound spots were visualized under the UV light at 254 and 365 nm and then derivatized with a combination methanol and sulfuric acid (9:1 *v*/*v*) to enhance visualization. The fractions that had a similar profile from TLC analysis were bulked together, leading to 57 major fractions from the ethanol extract. These major fractions were dried in pre-weighed beakers and their masses were determined as well.

#### 4.3.1. Bulking of Fractions from Ethanol Extract Column

Similar to bulking method used during purification of the hexane extract, contents of test tubes (TT 1-6) were combined to afford major fraction A. The other major fractions obtained from the chromatography of the ethanol extract were B(TT 7-13), C(TT14-28), D(TT29-70), E(TT71-80), F(TT81-90), G(TT91-100), H(TT101-110), I(TT111-130), J(1TT31-150), K(TT151-160), L(TT161-166), M(TT167-175), N(TT176-185), O(TT186-200), P(TT201-214), Q(TT215-228), R(TT229-242), S(TT244-252), T(TT253-260), U(TT261-280), V(TT281-380), W(TT381-412), X(TT413-436), Y(TT437-449), Z(TT450-460), AA(TT461-470), BB(TT471-490), CC(TT491-510), DD(TT511-550), EE(TT551-580), FF(TT581-590), GG(TT591-600), HH(TT601-610), II(TT611-616), JJ(TT617-620), KK(TT621-628), LL(TT629-632), MM(TT633-636), NN(TT637-650), OO(TT651-672), PP(TT673-680), QQ(TT681-685), RR(TT686-688), SS(TT689-720), TT(TT721-754), UU(TT755-788), VV(TT789-810), WWTT(811-812), XX(TT813-818), YY(TT819-840), ZZ(TT841-848), AAA(TT849-860), BBB(TT861-865), CCC(TT866-870), DDD(TT871-875), and EEE(TT876-880). The test tube portions were bulked in a round bottom flask and concentrated under pressure to afford a dry mass labeled sub-fraction E1(S-U), E2(V-VW), E3(A–R), E4(MM), E5(NN), and E6(OO-EEE). Whereas most of the sub-fractions revealed more than a spot on TLC analyses and were discarded, sub-fractions E1 and E3 had single compact spots, thus indicating good purity compounds. The two compounds were further characterized to examine their purity.

#### 4.3.2. Bacterial Culture and Maintenance

The pathogens used were *Staphylococcus aureus* (ATCC 25923), *Escherichia coli* (ATCC 10536), *Pseudomonas aeruginosa* (ATCC 9721), *Neisseria gonorrhoeae* (ATCC 49981), and *Streptococcus pyogenes* (ATCC 19615), purchased from Sigma-Aldrich (Pretoria, South Africa) at University of Pretoria. They were selected based on their pathogenicity, clinical relevance and the literature and were used to evaluate antimicrobial activity of crude extracts and proposed pure compounds isolated from *M. oleifera* Lam leaves’ hexane extract as well as ethanol extract. Stock bacterial cultures were sub-cultured into freshly prepared Mueller–Hinton agar (MHA) and incubated at 37 °C for 18–24 h to produce fresh bacterial culture. However, *Neisseria gonorrhoeae* (ATCC 49981) was incubated in a jar with carbon dioxide at the same temperature. To keep the bacterial strains alive, glycerol stock cultures of each organism were prepared and stored at 80 °C until needed.

#### 4.3.3. Preparation of Inoculum

Bacterial colonies were transferred to sterile Mueller–Hinton broth (MHB) which was stored in the refrigerator before the experiment. This was performed to standardize the overnight cultures by diluting with MHB until all bacteria had an absorbance (OD600 nm) of 0.08–0.1. The same protocol was applied throughout the experiments for culture preparation.

### 4.4. TLC Bioautography

The antimicrobial screening of *M. oleifera* Lam extracts and isolates were tested using direct bioautography evaluation on a TLC plate (10 × 20 cm, 0.25 mm thickness, silica gel G 60 F254, Merck, Darmstadt, Germany). The extracts and isolated compounds were spotted on ten plates (for five bacteria and executed in duplicate) at a concentration of 5 mg/mL and left to dry for five days at room temperature. The plates were developed with chloroform: ethyl acetate: formic acid (4:3:1 *v/v/v*) and hexane: ethyl acetate (9.5:0.5 *v*/*v*) solvent mixtures in a glass chamber for ethanol extracts and isolates and hexane extracts and isolates, respectively. This process was carried out in a fume hood cabinet (Vivid Air, manufacturer and suppliers of clean air equipment). The developed air-dried plates were placed in sterile Petri dishes, and a 10 μL inoculum of all selected bacteria was poured into every 5 mL of melted soft agar (composed of 1.3 g bacteriology agar, 2 g tryptone, 1 g sodium chloride (NaCl), and 200 mL distilled water (dH_2_O) and then distributed over the plates. The plates were incubated at 32 °C for 24 h after the nutrient agar had solidified. Following that, the bioautograms were sprayed with a solution of 0.2 mg/mL INT. On the TLC plate, inhibition zones appeared as distinct spots on a purple background.

#### 4.4.1. In Vitro Qualitative and Quantitative Antioxidant Assay

A solution of 2,2-diphenyl-1-picryl-hydrazyl (DPPH) was used to qualitatively test the antioxidant activity of each extract. Following the development of each plant extract solution on TLC, which was allowed to dry, the plate was sprayed with 0.2% DPPH in methanol solution. As for the quantitative antioxidant potentials of the extracts and isolated compounds the underlisted protocols were followed.

#### 4.4.2. DPPH Antioxidant Activity Assay

This was accomplished by modifying a method described by (Moyo et al., 2012) [38] and (Olivier et al., 2017) [39]. To that end, non-polar n-hexane extract, polar ethanol extract, and isolated compounds (compound E3 and Ra), were prepared in a range of concentrations (0.2 to 1.0 mg/mL). In a test tube, 1.0 mL of the extract solution was mixed with 1.0 mL of a DPPH solution containing a concentration of 0.2 mg/mL. The contents of the test tube were vortexed to thoroughly mix them together before being placed in a dark cardboard for 30 min. An aliquot of 200 μL was transferred into a 96-well plate. Thereafter, the absorbance of the various concentrations was measured spectrophotometrically at 517 nm using a 96-well microplate-reader spectrophotometer (SprectraMax^®^, Molecular Devices, San Jose, CA, USA). As reference standards, the same concentrations of ascorbic acid and butylated hydroxyl toluene (BHT) were used. The extracts’ percentage radical scavenging activity was calculated using the equation below.
$$\%\;\mathrm{DPPH}\;\mathrm{radical}\;\mathrm{scavenging}\;\mathrm{activity}=\frac{A_{\mathrm{control}}-A_{\mathrm{sample}}}{A_{\mathrm{control}}}\times 100$$
where A_sample_ = absorbance of the sample, A_control_ = absorbance of the negative control.

#### 4.4.3. Hydrogen Peroxide Activity Assay

The method described by Olivier et al. (2017) [25] was followed with minor modifications to assess the hydrogen peroxide (H_2_O_2_) scavenging potential of *M. oleifera* extracts and isolated compounds. Different concentrations (0.2 to 1.0 mg/mL) of the extracts and isolated compounds were prepared and 1 mL of each was transferred into a test tube. A volume of 2 mL hydrogen peroxide (20 mM) prepared in a phosphate buffer saline (pH 7.2) was mixed with 1.0 mL of each of the extracts and isolated compounds from the stock solutions. The reaction was thoroughly mixed with a vortex at 3000 rpm and incubated for 10 min at room temperature. An amount of 200 μL of each mixture was transferred into a 96-well microtiter plate and the absorbance was measured at 560 nm with a spectrophotometer. As positive control standards, BHT and ascorbic acid were used. The extracts’ ability to scavenge H_2_O_2_ was calculated using the equation below.
$$H_{2}O_{2}\;\mathrm{scavenging}\;\mathrm{activity}\;\%=\frac{A_{0}-A_{s}}{A_{0}}\times 100$$
where A_0_ = absorbance of negative control and A_S_ = absorbance of sample.

#### 4.4.4. Reducing Power Activity Assay

The method used in this study was adapted from (Moyo et al., 2012) [38] with minor modifications. The various extracts and compounds were redissolved in the solvents from which they were extracted. Following that, various concentrations ranging from 0.2 to 1.0 mg/mL were prepared. In a test tube, 2.5 mL of 0.2 M potassium phosphate buffer (pH 7.2) and 2.5 mL of 1% (*w*/*v*) potassium ferricyanide (K3Fe(CN)6) were mixed by means of vortexing at 3000 rpm. After mixing the contents in the test tube, they were incubated for 20 min at 50 °C. Following that, 2.5 mL trichloroacetic acid (TCA) (10% *w*/*v*) was added to the mixture and centrifuged for 10 min at 3000 rpm. An amount of 2.5 mL of the solution’s upper layer was mixed with 2.5 mL of distilled water and 0.5 mL of ferric chloride (FeCl_3_) (0.1% *w*/*v*). The absorbance of the mixture was measured using a spectrophotometer at 700 nm against blank. The procedure was repeated for the reference standards ascorbic acid and BHT. The extracts’ percentage reducing power was calculated using the following equation:$$\mathrm{Reducing}\;\mathrm{power}\;\mathrm{activity}\;\%=\frac{A_{0}-A_{s}}{A_{0}}\times 100$$
where A_0_ = absorbance of negative control, A_S_ = absorbance of sample

#### 4.4.5. Statistical Analysis

A *t*-test statistical analysis was performed to compare the probability (*p*-value) of significance between the three assays used to investigate the antioxidant potential of the extracts and isolated compounds in Microsoft Excel^®^.

## 5. Conclusions

The study has successfully investigated and established the antioxidant and antimicrobial potential of South African *Moringa oleifera* Lam leaves extracts and three isolated compounds. Even though the study found no antimicrobial activity against *Staphylococcus aureus* (ATCC 25923), *Escherichia coli* (ATCC 10536), *Pseudomonas aeruginosa* (ATCC 9721), *Neisseria gonorrhoeae* (ATCC 49981), and *Streptococcus pyogenes* (ATCC 49981) (ATCC 19615), our results are among the very few to the best of our knowledge that will inform local farmers and other stakeholders on the possibility of using South African-based *Moringa* oleifera leave products that may mitigate oxidative stress and related diseases.

## Figures and Tables

**Figure 1 molecules-28-00899-f001:**
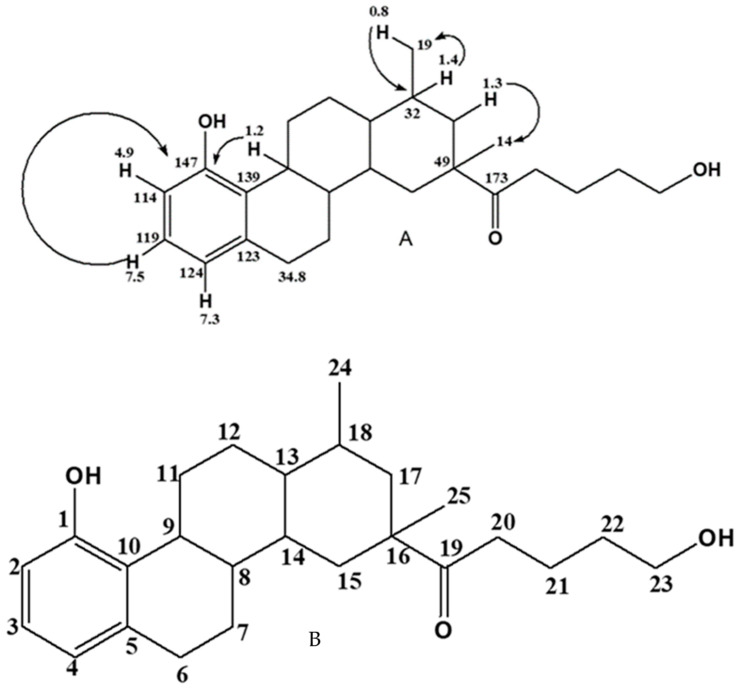
HMBC correlations that linked the various moieties (**A**) to form the proposed skeletal structure (**B**) of E1.

**Figure 2 molecules-28-00899-f002:**
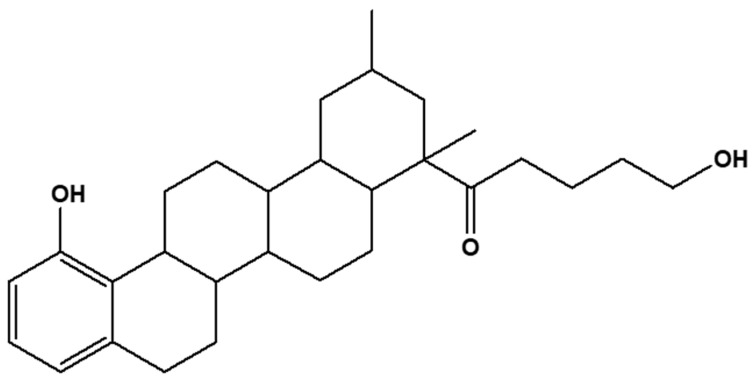
Structure of E1 (10-hydroxy-1,3-dimethylchrysen-3-yl)-5-hydroxypentan-1-one).

**Figure 3 molecules-28-00899-f003:**
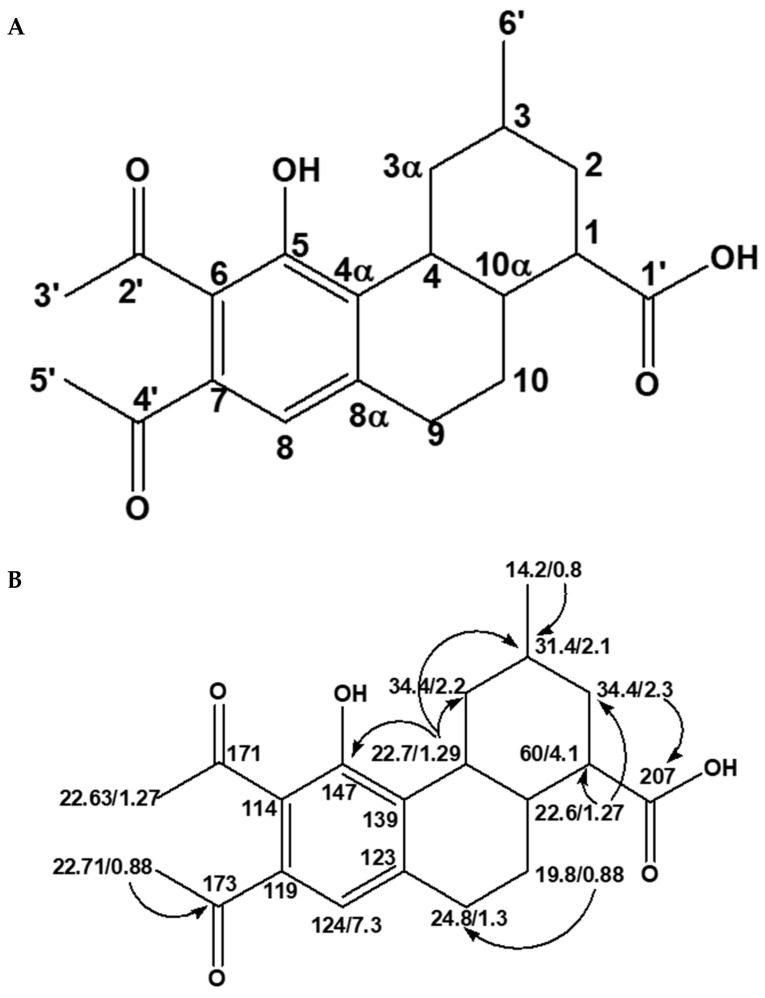
Skeletal structure (**A**) and HMBC correlations (**B**) of E3.

**Figure 4 molecules-28-00899-f004:**
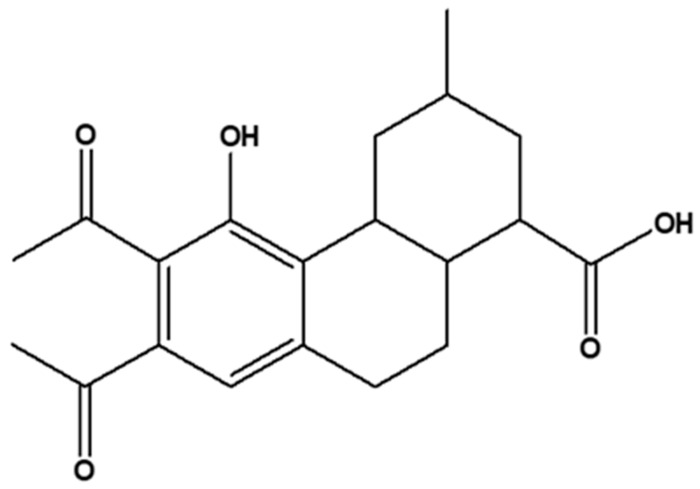
Structure of compound E3 (6,7-dipropanone-5-hydroxyphenyl-3-methylphenanthrene-1-carboxylic acid).

**Figure 5 molecules-28-00899-f005:**
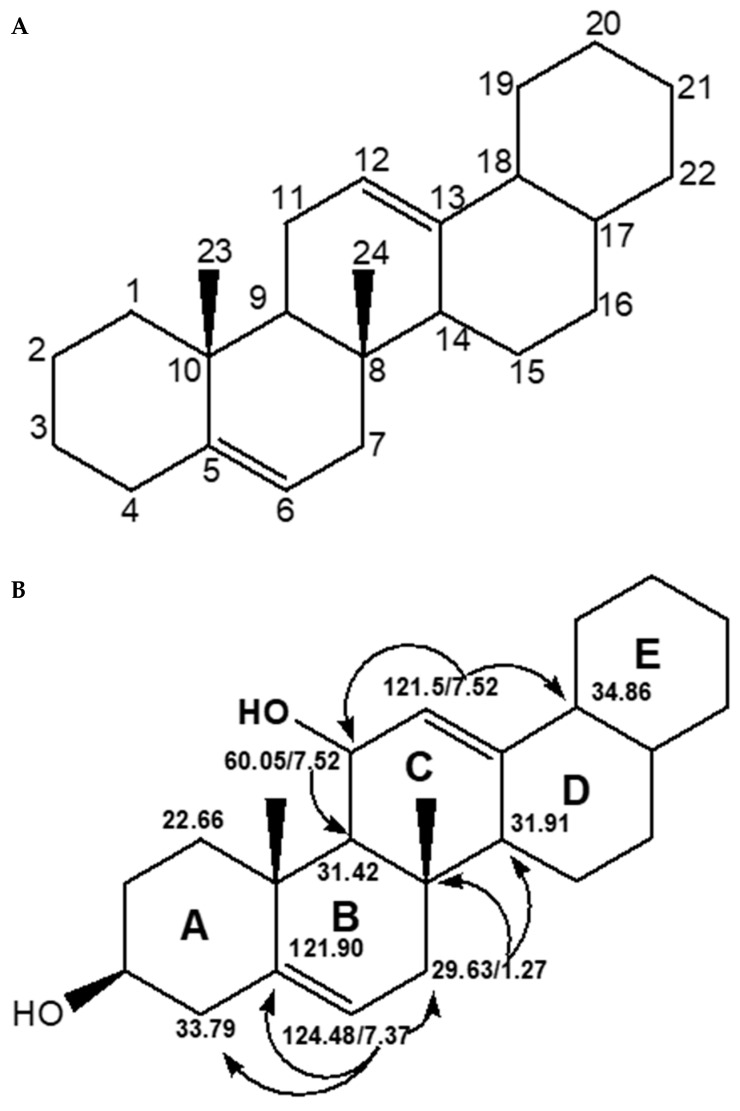
Skeletal structure of (**A**) HMBC connectivity of the different moieties (**B**) of Ra isolated from the hexane extract of *M. oleifera* leaves.

**Figure 6 molecules-28-00899-f006:**
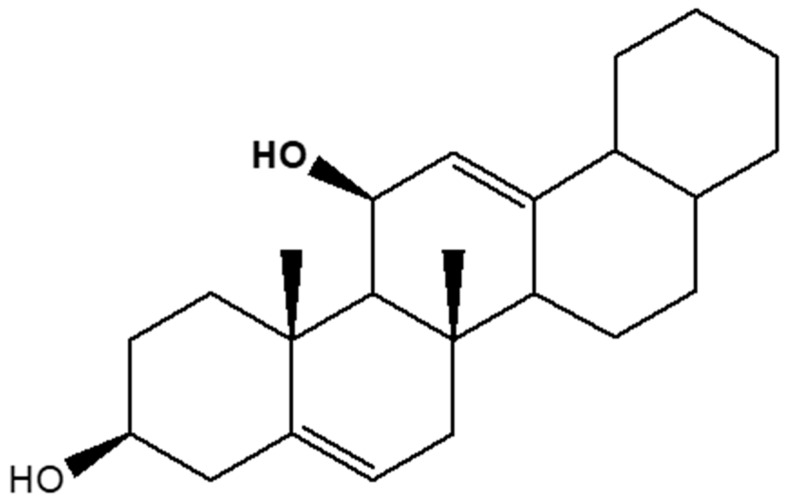
Proposed structure of Ra isolated from the hexane extract of *M. oleifera* leaves and elucidated as hexademethylated 3β,11β-dihydroxyfriedelane.

**Figure 7 molecules-28-00899-f007:**
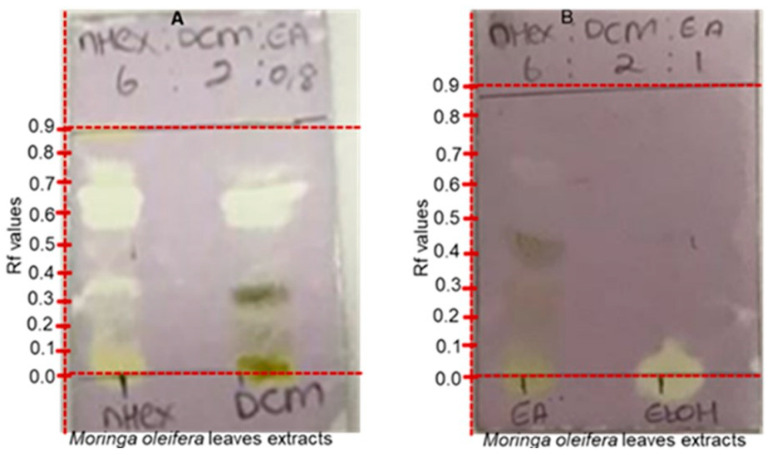
Developed TLC plates of hexane, dichloromethane extracts (**A**), and ethyl acetate and ethanol extracts (**B**). Plates were stained with 10% DPPH solution and visualized under visible light.

**Figure 8 molecules-28-00899-f008:**
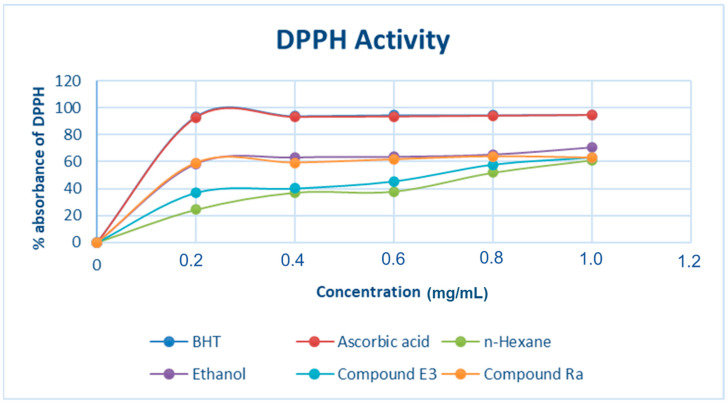
DPPH radical scavenging activity of *M. oleifera* extracts and pure compounds. Data are presented as the percentage of DPPH radical scavenging. Each value is expressed as mean ± standard deviation (*n* = 3).

**Figure 9 molecules-28-00899-f009:**
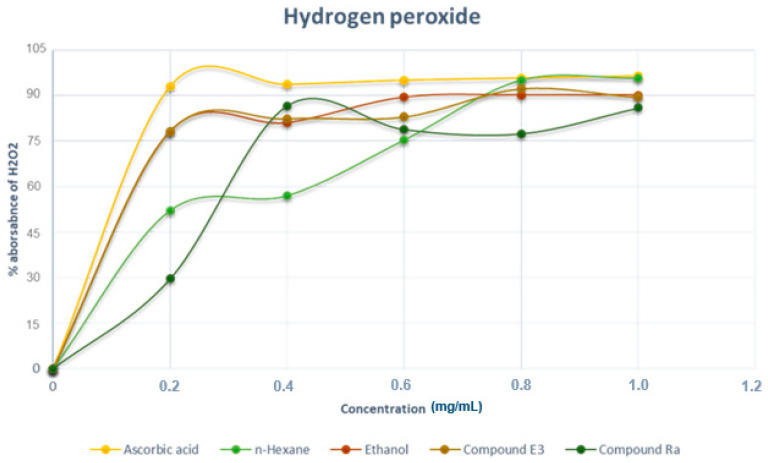
Scavenging activity (%) of H_2_O_2_ of *M. oleifera* extracts and pure compounds. Data are presented as the percentage of H_2_O_2_ radical scavenging. Each value is expressed as mean ± standard deviation (*n* = 3).

**Figure 10 molecules-28-00899-f010:**
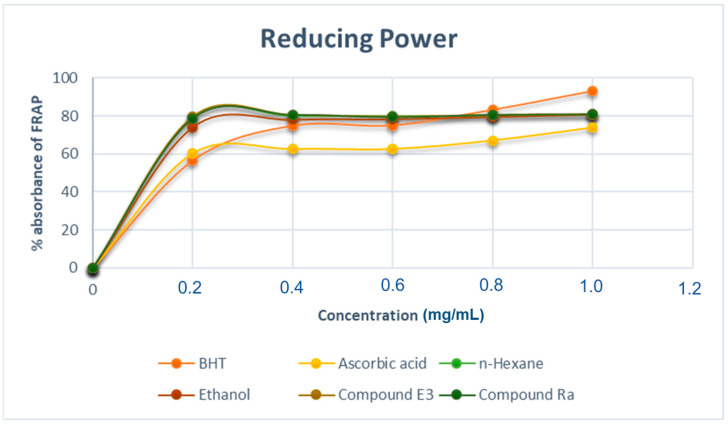
% Reducing power of *M. oleifera* extracts and pure compounds. The results are presented as the percentage reducing power free radical scavenging activity. Each value is expressed as mean ± standard deviation (*n* = 3).

**Table 1 molecules-28-00899-t001:** Proton and carbon-13 signals and the multiplicity of E1.

Position	C-δ (ppm)	H-δ (ppm), *J* (Hz)
1	147	Cq
2	114	7.14 (H, d, *J* = 8.4)
3	119	7.54 (H, dd, 11.1, 3.1)
4	124	7.37 (H, d, *J =* 14.0)
5	123	Cq
6	34	2.29 (2H, m)
7	24	1.69 (2H, dt, *J =* 11.4, 7.4)
8	39	1.30 (H, m)
9	28	2.02 (H, m)
10	139	Cq
11	27	2.09 (2H, dd, *J =* 11.3, 4.2)
12	31	2.36 (2H, m)
13	50	3.51 (H,
14	29.6	1.30 (H, m)
15	30	2.10 (2H, t, *J* = 5.7)
16	49	Cq
17	37	1.32 (2H, m)
18	32	1.44 (H, dd, *J* = 12.1, 2.9)
19	173	Cq
20	33	2.36 (2H, dd, *J =* 11.3, 3.2)
21	22	0.89 (2H, m)
22	29.5	1.30 (2H, m)
23	60	4.17 (2H, dd, *J =* 11.5, 3.0)
24	19	0.89 (3H, s)
25	14	0.90 (3H, s)

Cq = quarternary carbons.

**Table 2 molecules-28-00899-t002:** The carbon-12 and proton (H-1) signals for E3.

Position	C-δ (ppm)	H-δ (ppm), *J* (Hz)
1	60.16	4.13 (q, H, *J* = 7.2)
2	39.89	2.02 (dd, 2H, *J* = 11.8, 7.0),
3	28.96	1.63 (m, H)
3α	34.42	2.33 (dd, 2H, *J* = 11.7, 7.2),
4	31.44	2.19 (m, H)
4α	139.03	Cq
5	147.09	Cq
6	114.08	Cq
7	119.09	Cq
8	124.47	7.38 (s, H)
8α	123.99	Cq
9	30.94	1.5 (dd, 2H, *J* = 13.3, 3.6)
10	24.81	1.30 (m, 2H)
10α	29.68	2.02 (m, H, *J* = 10.8, 7.6)
1′	207.02	Cq
2′	171.00	Cq
3′	22.63	1.27 (s, 3H)
4′	173.98	Cq
5′	22.71	0.88 (s, 3H),
6′	14.21	0.87 (s, 3H)

Cq = quaternary carbons.

**Table 3 molecules-28-00899-t003:** The carbon-12 and proton (H-1) signals for Ra.

Position	C-δ (ppm)	H-δ (ppm), *J* (Hz)
1	22.66	1.36 (2H, dd, *J =* 10, 4)
2	29.33	1.14 (2H, td, *J* = 12, 4)
3	68.32	3.87 (H, m)
4	34.49	1.23 (2H, d, *J =* 8)
5	119.90	Cq
6	124.48	7.37 (H, m)
7	29.63	1.30 (2H, m)
8	29.67	Cq
9	31.42	2.02 (H, m)
10	30.21	Cq
11	68.03	3.59 (H, td, *J* = 12, 4)
12	114.03	7.51 (H, m)
13	123.96	Cq
14	31.91	2.06 (H, t, *J=* 8)
15	29.49	1.31 (2H, td, *J =* 8, 4)
16	28.94	1.15 (2H, dt, *J =* 16, 4)
17	34.52	1.30 (H, m)
18	34.86	2.00 (H, m)
19	29.14	1.99 (2H, m)
20	28.55	1.29 (2H, m)
21	27.07	1.13 (2H, tt, *J* = 16, 4)
22	30.02	1.26 (2H, td, *J* = 8, 4)
23	18.81	1.24 (3H, s)
24	14.04	0.93 (3H, s)

Cq = quaternary carbons.

**Table 4 molecules-28-00899-t004:** IC_50_ values (mg/mL) of *M. oleifera* extracts and pure compounds in DPPH scavenging, hydrogen peroxide, and reducing power assays.

Analyte	DPPH IC_50_ (mg/mL)	H_2_O_2_ IC_50_ (mg/mL)	FRAP IC_50_ (mg/mL)
BHT	0.08	Nd	0.323
Ascorbic acid	0.88	0.443	0.421
n-Hexane	0.761	0.639	0.211
Ethanol	0.435	0.541	0.249
Compound E3	0.671	0.559	0.208
Compound Ra	0.475	0.689	0.213

Nd = Not determined. Each value is expressed as mean ± standard deviation (*n* = 3).

## Data Availability

Not applicable.

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
