# Peer review of "Antioxidant and Antimicrobial Evaluations of Moringa oleifera Lam Leaves Extract and Isolated Compounds"

_molecules, 2023, doi:10.3390/molecules28020899_

Round 1
Reviewer 1 Report
Dear Authors,
I reviewed your manuscript submitted to the Molecules Journal where you address the potential antioxidant and antimicrobial effects of Moringa oleifera Lam leaves extracts and Isolated 3 Compounds.
I have some suggestions that can help the presentation of obtained results.
Major:
1. The title needs a revision
2. Please eliminate the Simple summary section.
3. Revision of the abstract.
4. Lines 44, and 50 need English corrections.
5. The statement ”One of the major 56 public health concerns worldwide is of infectious diseases caused by pathogens such as 57 Staphylococcus aureus, Escherichia coli, Pseudomonas aeruginosa, Neisseria gonorrhoeae, and 58 Streptococcus pyogenes” needs to be backed up by more than 1 reference.
6. Revision of lines 103, 111
7. Sections 2.1 and 2.2 need to be combined.
8. Section 2.5 needs to have subsections, not the 2.6 section. Please enumerate in section 2.5 all the antioxidant methods used for assessment, and then detail them in the subsections.
9. Please provide the HPLC/ UPLC chromatograms obtained. As well as the NMR spectra.
10. Please provide a higher-resolution picture for figure 7.
11. Is there any statistical analysis and relevance of results from Table 1? If so, please provide it.
12. The Discussion section needs to be expanded as it focuses only on the antioxidant activity.
13. The Conclusions section needs to be revised entirely as it focuses only on the antimicrobial effect.
Due to the fact that the obtained extract presented no antimicrobial activity, the introduction section needs to be reevaluated and modified accordingly. With a focus on the lack of antimicrobial assessments, a lack of understanding regarding the compound-effect relation, etc.
This manuscript also needs a thorough run-through and English proofreading.
Good luck!
Author Response
Dear Reviewer,Please find attached the rebuttal and the revised manuscript molecules-2124919.
Thanks, and kind regards. Dr Bassey

Reviewer 2 Report
Manuscript Number: Molecules-2124919
Title: Qualitative and Quantitative Antioxidants and Antimicrobial Activity of Moringa oleifera Lam leaves extracts and Isolated Compounds
The piece of work is timely, interesting and generated some useful information on valuable pharmacological properties such as anti-asthmatic, anti-diabetic, anti-in- flammatory, anti-infertility, anti-cancer, anti-microbial and antioxidant of Moringa oleifera. Overall, the manuscript was written well to explain the objectives. However, I have some minor issues that are to be addressed before the article being accepted. These are as follows:
Minor Comments:
1. Please improve the discussion part
2. The conclusion should be more specific in terms of results.
Author Response

(The authors gave the same response as above.)

Reviewer 3 Report
The authors have evaluated the qualitative and quantitative antioxidants and antimicrobial activity of Moringa oleifera Lam leaves and their isolated compounds. The prepared extracts and isolated compounds were characterized well and evaluated for antioxidant and antimicrobial activities. Overall, the work is interesting and will be beneficial for the fellow researchers. I will recommend it after major revisions:
Abstract: The quantitative results are missing. The authors are suggested to include quantitative results in order to enhance the readability of the manuscript.
Symbols, units, subscripts and superscripts: The authors are advised to present all units in SI system and there should be a space between the physical quantity and unit. All the symbols should be italics and their subscripts or superscripts should be non-italics throughout the manuscript.
Please rearrange the sections of manuscript according to journal style i.e., 1. Introduction, 2. Results, 3. Discussion, 4. Materials and Methods, and 5. Conclusions.
Introduction: The importance of antioxidant and antimicrobial part is quite swallowed. The authors are advised to add some literature about the importance of antioxidant and antimicrobial activity of natural products. You can consult the following articles to make this manuscript more useful to the readers:
Acta Physiol. Plant. 37: E253 (2015); J. Food Sci. Technol. 57:1191-1204 (2020); Nat. Prod. Res. 26: 460-465 (2012); Life Sci. 165: 1-8 (2016); ACS Omega 5: 6461-6471 (2020).
Figures 8-10: Kindly include the number of replicates and error bars in the graphs.
Tables: Kindly include the number of replicates in all data tables.
Discussion: Please compare your results with previous studies and mention clearly how your work is important in comparison to already been reported.
Authors are advised to include the main limitation of work at the end of discussion section and just before the conclusion.
Avoid abbreviations before giving their explanation in the abstract, text, table, and figure.
Conclusion: The conclusion should be concise and to the point indicating the application of the work.
Author Response

(The authors gave the same response as above.)

Reviewer 4 Report
Reviewer’s comments
This research was evaluating the free radical scavenging ability of Moringa oleifera ethanolic and hexane extracts and pure compounds of M. oleifera Lam (hexane, ethanol, compound E3 and compound Ra) against reactive oxygen species, as well as their reducing power and antimicrobial activities. From the results, the degree of novelty on antioxidant and antimicrobial of this plant is apparently a very common. However, it is possible that certain findings related to the purified compounds of the extracts can be of interest.
-Please carefully check the Instructions for Authors from the journal. Research manuscript sections should be revised to Introduction, Results, Discussion, Materials and Methods, and Conclusions.
- Introduction and Discussion sections are quite plain, not in deep. It is recommended to tone up these sections.
-The conclusion has to improve, the conclusion section should conclude the study and the further study base on your results, and should not have a reference.
Introduction
-This research study on Antioxidants and Antimicrobial Activity of Moringa oleifera, but the information about M. oleifera is very less in the introduction. Please add more information of M. oleifera.
-Line 50: Essential oils are not phytochemical, therefore delete these words. Alkaloids change to alkaloids.
-Line 111: the authors mentioned “satndard protocols”, what is “satndard protocols” please, briefly describe or reference.
-Line 115: What is DCM mean, abbreviations should be defined the first time.
-Line 322,379: HPLC chromatogram of compounds should be present at least in the Supplementary Materials.
-In line 546: Antimicrobial Assay
-As the results, the extracts and compounds Ra and E3 did not have antimicrobial effects. Anyway, the highest concentration used in the experiment should be indicated and MIC values of each extract or compound should be present.
-Line 547: In this section described the antimicrobial results, therefore “antioxidant” should be deleted from this sentence.
Material and methods
-To increase the value of the study, statistical analysis should be performed to compare the significance of the antioxidant activity of each treatment in Table 1.
-Unwanted spacing and typo mistakes throughout the manuscript. Need to be checked and corrected carefully.
- Please carefully check that Supplementary Fig 1-5 was mentioned in the main text body
Please carefully check as follow
-ml change to mL
-minutes change to min
-hour change to h
Author Response

(The authors gave the same response as above.)

Round 2
Reviewer 1 Report
Dear Authors,
Congratulations! I can now recommend your manuscript for publication.
Reviewer 3 Report
I am satisfied with the author's answer and agree to accept it.